# Picosecond Laser-Induced Bump Formation on Coated Glass for Smart Window Manufacturing

Savely Ioffe [1,2,*] , Andrey Petrov [1,2,3] and Grigory Mikhailovsky [1,2]

1. Nordlase Ltd., 198205 St. Petersburg, Russia
2. The World-Class Research Center "Advanced Digital Technologies", Peter the Great St. Petersburg Polytechnic University, 195251 St. Petersburg, Russia
3. School of Physics and Engineering, Saint Petersburg National Research University of Information Technologies, Mechanics and Optics (ITMO University), 197101 St. Petersburg, Russia
* Correspondence: ioffe_sava@mail.ru

**Abstract:** We report a study on the process of the formation of bubble-like structures on a coated glass surface using 50 ps laser pulses. The high-intensity interaction of laser radiation on the film–glass interface allowed us to develop a process for efficient glass bump formation. The high peak energy of the picosecond pulses has allowed us to merge the processes of coating evaporation and bubble growth into one. A parameter window was established within which efficient bump formation can be achieved. Well-defined spherical structures with a height up to 60 μm and a diameter up to 250 μm were obtained at pulse energy $E_{pulse} = 2.5 \div 4$ μJ and laser fluence $F = 2.5\text{–}0.41$ J/cm$^2$). The key aspects of the bump formation process were studied and are explained.

**Keywords:** picosecond pulses; laser machining; bump formation; glass swelling





## 1. Introduction

Femto- and picosecond laser systems are receiving significant attention for their unique interaction with matter. The ultrashort pulse duration produces an exceptionally high peak power (>1 MW) at a comparatively low average power. The high peak power as well as the short pulse duration significantly changes the nature of laser processing. When the pulse duration is shorter than the characteristic times of the thermal process, such as electron–phonon/phonon–phonon relaxation, it allows for the thermal energy to be concentrated in a small area. This leads to a higher level of precision in the process. In this scenario, the extent of energy penetration is primarily influenced by the optical depth of penetration as opposed to thermal diffusion processes. A high peak power increases the probability of multiphoton ionization, which makes it possible to process refractory, heat-resistant, and even transparent materials. These remarkable properties of ultrafast laser systems have made it an ultimate tool for a wide range of applications: material modification [1], micromachining [2], ophthalmology [3], and spectroscopy [4].

Investigations of the ultrashort laser machining of glass are currently underway and there are still many challenges regarding its application, mainly due to glass brittleness and transparency. Optical glass demonstrates beneficial properties such as high chemical stability, thermal resistance, dispersion, and wavelength range, on top of that, silica glass is widely available; therefore, it is utilized in various technical products. The interaction between glass and ultrashort laser irradiation goes as follows: when intense femtosecond laser pulses are focused within a transparent glass, the intensity at the focal point is extremely high and can induce nonlinear absorption such as multiphoton and/or tunnel absorption. The energy from the laser melts the material at the focal volume, leading to structural changes as it resolidifies around the focal point. By focusing femtosecond laser pulses on the inner interfaces of transparent materials, a liquid pool is formed through localized melting caused by nonlinear absorption. This process is crucial in the welding of

the glass to glass. The first demonstration of welding glass by ultrashort laser pulses was reported by Takayuki Tamaki et al. [5] in 2005. In their research, silica glass was exposed to 800 nm, 130 fs laser pulses at the repetition frequency of 1 kHz. Up until now, nearly the same conditions have been used in different micro-machining processes, such as gap bridging [6], drilling [7], multi-material joining [8], and 3D patterning [9].

The interaction of ultrashort laser pulses and glass excites intriguing phenomena such as the formation of microbubbles/voids in the bulk [10] or bumps on the surface [11,12] of the exposed material. The main idea behind the bump formation on a surface using ultrashort laser pulses is local heating concentrated on a small area above the glass transition temperature. Then, when it cools down, the glass remains in its non-equilibrium state, creating bump-like structures on the surface of the glass, metallic films [13,14], and even polymers [15]. Bump generation depends heavily on the material's properties, which are the composition of the glass, its density, the glass transition temperature, and Young's modulus [16], and laser settings such as the exposure time, repetition rate, high peak power, and pulse duration. Also, the size and shape of bumps can be altered by changing the laser focus inside the material [17]. The process of bump formation serves as an efficient technique for creating 3D patterns on the surface of glass machining areas. This technology is widely utilized in photonics and surface texturing [18]. One of the particularly promising applications is the manufacturing of smart windows [18–20]. Such windows are widely used in construction and architecture, not only for aesthetic and constructive properties but more importantly for energy saving. Windows are the most efficient channel for thermal energy loss. The structure of the smart window is schematically represented in Figure 1.

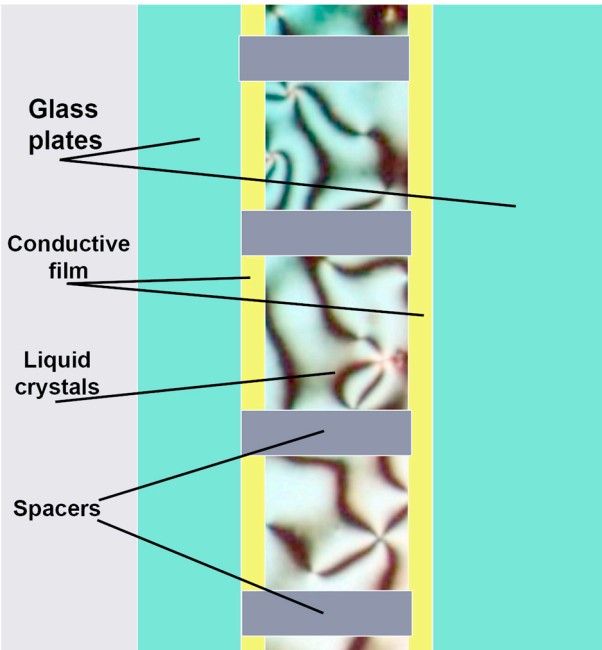

**Figure 1.** Smart window layout.

Smart glasses have become widespread in architecture and construction in the last decade. The variable transparency of such glasses makes it possible to solve a number of problems. In addition to aesthetic purposes, smart glasses have a wide range of functional properties, such as optimizing the working space in offices (the ability to isolate individual rooms if necessary), ensuring privacy in medical wards, and, most importantly, reducing the cost of space heating. The ability to regulate the transparency of windows depending on the temperature (thermochromic) or manually by the user or the program, can significantly reduce heat loss in rooms with such windows. The main elements are conductive/thermochromic coating, liquid crystal, and spacers.

Currently, various technologies are used for the production of spacers, for example, polymer printing; however, in this case, a significant deviation in dimensions during polymerization can be observed, and this process is also quite time-consuming. Conductive films of tungsten, vanadium, cerium dioxide, and vanadium dioxide [21] deposited by physical and chemical vapor deposition techniques are widely used as thermochromic coatings in smart window production. The formation of bumps on the metallic thin layers has been reported, though mostly for noble metals, such as Au [13,14], Cu [22], and Ag [23]; however, authors have obtained highly precise deposited and controllable arrangements of submicrometer bumps and jets. Femtosecond laser pulses are commonly used for bump generation due to their higher precision and accuracy compared to picosecond pulse durations; nevertheless, the low speed and size of the resulting bumps are not fully suitable for a cost-efficient manufacturing process on a large area required for the production of smart windows. The main nonlinear mechanism for the interaction of femtosecond radiation with glass is nonlinear absorption [24] and propagation (filamentation) [25]. Those effects are caused by the high peak intensity of the tightly focused femtosecond laser beam. At the same time, the longer interaction time of picosecond pulses enables such nonlinear effects as avalanche ionization, phase explosion, etc. [26,27]. In general, for picosecond pulses, the thermal mechanism is more pronounced, which makes it more preferable for bump formation, since thermal-driven mechanisms such as viscous flow are involved in the bubble formation process. Moreover, in the case of processing with femtosecond pulses, the focusing conditions have a crucial impact on the nature of the interaction. D. Kawamura et al. have demonstrated that a small focus shift (1000 nm) leads to a significant reduction in the structure height [28]. For the stable process of bump formation, extremely precise control over the focus position is required in the case of femtosecond laser machining, which limits it's applicability for industrial-grade processes. In this regard, the utilization of picosecond lasers permits much more opportunities. An increased pulse duration can result in a higher average power output compared to femtosecond lasers. Hence, their faster scanning speed is due to their ability to deliver a higher average power, resulting in quicker material machining on larger processing areas, preserving a relatively high precision and high localization extent [29]. Furthermore, for economical considerations, using picosecond lasers generally allows for the achievement of a more streamlined and affordable setup compared to femtosecond lasers [30]. Due to the complexity of manufacturing, femtosecond lasers are, on average, more than twice as expensive as picosecond systems with similar parameters. As a result, picosecond lasers offer a better balance between precision and processing speed. Therefore, using picosecond laser induction for spacer formation is a promising alternative method. The scanning laser system can greatly enhance and streamline the formation process of spacers by combining high speed and precision with the ability of bump formation on the conductive coatings on glass. The typical thickness of a liquid crystal layer is in the order of 20–50 μm. For the production of smart windows, it is necessary that the thickness of the liquid crystal layer lies in this range, since the increasing of the layer thickness leads to the deterioration of its optical characteristics (decreased transparency) and the uniformity of the layer, caused by the high concentration of air bubbles. On the contrary, when the thickness is too low, the process of filling the interplate space with liquid crystal becomes significantly prolonged, which can significantly increase the cost of the production process.

To improve glass micromachining, it is necessary to explore and understand the complex mechanism of ultrashort pulsed laser ablation. Picosecond laser-induced bump formation is rarely published, mostly because it is hard to achieve the efficient interaction of picosecond pulses with the transparent glass material; although, it has some advantages over the femtosecond pulse laser, such as a rapid scanning speed, higher average power output, and cost-effectiveness, which can be applied for the development of technology for spacers manufacturing for smart windows, glass surface modification [31,32], and microfluidic devices [33,34]. This paper delves into the formation of bump-like structures

on glass surfaces and conductive coatings deposited on the glass substrate by varying the laser focus planes and repetition rates.

## 2. Materials and Methods

Experiments were carried out with the help of the in-house-built picosecond laser with the following parameters: the fixed pulse duration was $50 \pm 5$ ps, the emission wavelength was 1064 nm, and the average power was up to 60 W, with pulse repetition rate from 0.5 kHz up to 10 MHz. The experimental laser set up consists of a focusing system based on the focusing lens (F = 50 mm), which provides a minimum spot diameter of $1/e^2 = 50$ µm. The beam shape is Gaussian and beam quality is $M^2 < 1.3$. The position of the focusing lens was controlled by a high-precision Z-drive with a vertical resolution up to 1 µm. Sample positioning was controlled with PI L-731 Precision stage, which provides a scanning speed of up to 200 mm/s and coordinate repeatability of 0.5 µm. Three specimens of $100 \times 100 \times 4$ mm of commercially available smart window glass were investigated: NC—non-coated sample of the glass substrate; FTO—sample with a conductive layer of tin–fluorine oxide; PVD—sample with a conductive layer of i/iM glass deposited by physical vapor deposition. The laser system has no built-in pulse picker, so for the selection of pulse number, a trigger system was used. The XY motion stage allowed us to irradiate the sample along a line with two trigger points—lasing on and lasing off. Trigger points were set so that the distance between them was equal to the minimum laser spot diameter—50 µm. In that case, every pulse was located within the spot area. By varying the linear motion speed, it was possible to control the number of pulses (exposition time) that irradiated the sample (Figure 2). The jitter of the laser system was measured: timing jitter was $-2$ ps, period jitter was 100 ps, and amplitude jitter was about 2%.

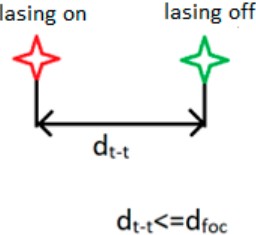

**Figure 2.** Scheme of point formation, $d_{t-t}$—trigger to trigger distance.

To study the effect of laser focus shift, the focusing lens was displaced relative to the sample surface, the starting point was settled at $-300$ µm, then, step-by-step, the laser irradiation was initiated each 100 µm from $-300$ µm up to 400 µm and results were recorded using optical microscope. To determine the influence of process parameters on the laser pulse-coated glass interaction, repetition rate (f), average power (P), exposition time ($\tau$) were varied. The focusing was performed manually, as follows: a polished metal plate was placed on the table, then, using a camera, the position of the focusing lens was adjusted to achieve the minimum diameter of the focal spot. After this, the thickness of the metal plate was measured with a micrometer; the resulting thickness value was then added to the Z-drive coordinate: $Z_{interface} = Z_f + h$, where $Z_{interface}$—Z-drive coordinate for interface focus (due to the small thickness of the coating, its thickness can be neglected), $Z_f$—focusing position, which corresponds to the best focusing conditions on the metal plate and $h$—thickness of metal plate.

In order to estimate the height of the formed structures, Mitutoyo digimatic micrometer was used alongside the method of contact profilometry, using a MARSURF PS 10 profilometer. This device is designed to measure surface roughness; therefore, it makes it possible to estimate the height difference along the established line. An optical microscope was used to determine diameter and shape of induced structures.

### 3. Results and Discussion

Firstly, a series of experiments was undertaken in order to study the process of bump formation on the glass surface without a conductive coating. In the experiments, the processing parameters (average power P, pulse frequency f, exposition time $\tau$, focus position) were varied in order to evaluate the processing window for efficient glass swelling. It was found that irradiating the glass with pulse energy $E_p > 3.0$ μJ (F > 0.31 J/cm$^2$), leading to the modification of the glass, happens in the volume of the sample rather than on the surface, at any focus position and applied power. The modified regions manifested as a dots comparable in size to the size of the focal spot (50 μm) and presumably consisted of locally recrystallized glass material. One can make an assumption that these regions represent a local change in the refractive index, which is typical for the processes of glass modification with ultrafast laser pulses[24]. A photomicrograph of the modified areas is presented on Figure 3.

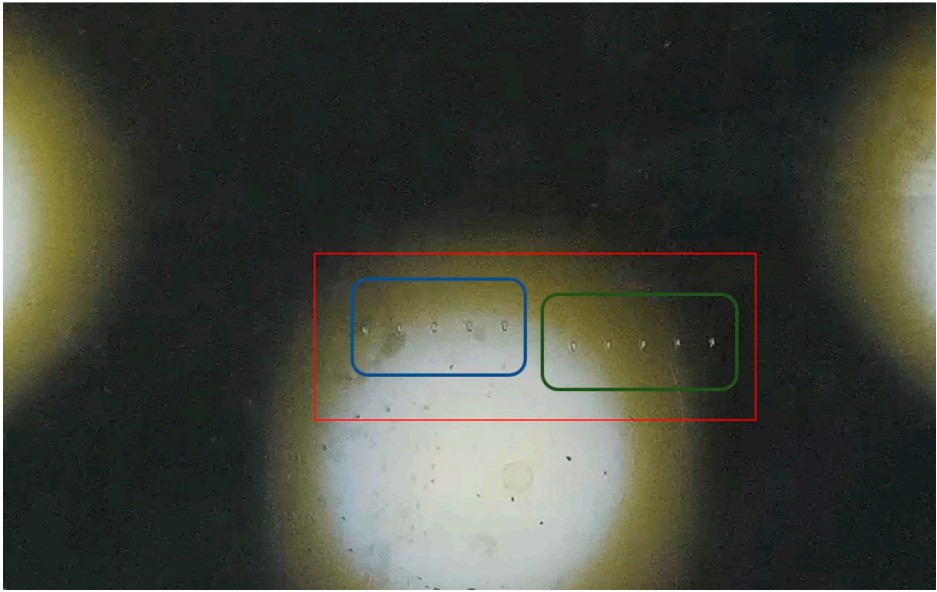

**Figure 3.** Altered areas in uncoated glass. Formation parameters are P = 40 W, f = 5 MHz, $\tau$ = 50 ms, F = 0.81 J/cm$^2$. Blue rectangle—front plate irradiation, green—back plate irradiation.

The formation of the altered microregions in the volume of the glass samples is attributed to the poor absorption of 1064 nm radiation in glass. The power density is not high enough to initiate the processes of laser radiation absorption on the surface; although, the high-peak power and high probability of multi-photon ionization leads to the formation of altered material areas. This is a remarkable result, but it is not applicable as a part of the smart glass technological process.

After it was found that the absorption of the uncoated glass was not high enough to initialize the process of bump formation, it was decided to continue the experiments with coated glass samples. An experiment was carried out using the following modes: P = 30 W, f = 10 MHz, $\tau$ = 50 ms and corresponding laser fluence F = 0.3 J/cm$^2$ were set to be constant, and the focus position was varied. It was found that at the pulse frequency f < 5 MHz, only the evaporation of the conductive layer takes place (Figure 4). Two experimental schemes were used: front plate irradiation and back plate irradiation (Figure 5).

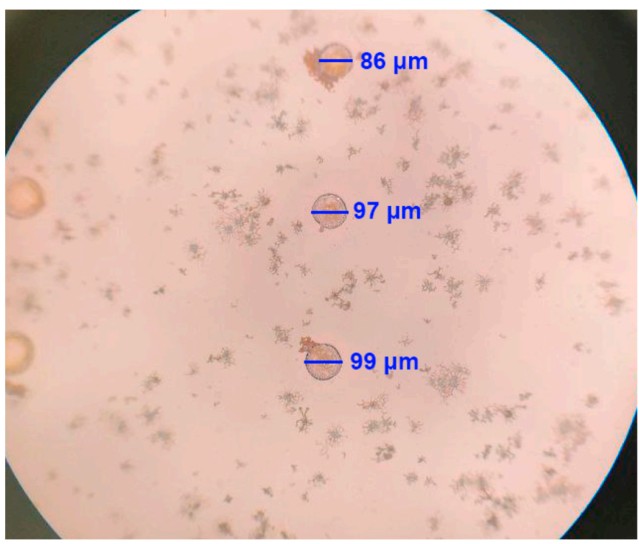

**Figure 4.** Evaporation of i-glass coating at f = 2 MHz mode.

## Back plate irradiation      Front plate irradiation

**Figure 5.** Surface irradiation scheme.

Surface modification and melting were observed within the irradiated glass–coating interface with the laser focused through the glass (back plate irradiation mode). Only the evaporation of the coating occurs in the front plate irradiation mode (Figure 6). The evaporated area has an elliptical shape, which corresponds to the formation direction. Such a shape is caused by the motion of the sample during irradiation. However, the irradiated areas on Figure 4 are purely circular. Thus, the elliptical shape may be attributed to the beam disturbance caused by passing through a volume of glass. Figures 7 and 8 demonstrate the structures formed on the sample surface in the back plate irradiation mode. The modified areas consist of the partially evaporated conductive coating film and resolidified glass. The film near the evaporation area has acquired a characteristic yellow color. The glass material melted due to exposure to the laser irradiation, forming a droplet on the surface, much larger than the size of the focal spot and the area of the film evaporation. One can observe characteristic features in the irradiated area, represented as small bubbles on the coating (red circles), which indicate the rapid evaporation of the film. Also, in the case of the formation of a droplet on the surface, the formation of microcracks can be observed, which may be attributed to the rapid local solidification of the material after laser exposition. The difference between the interaction during the direct irradiation of the film (front plate) and irradiation of the film–glass interface (back plate) may be due to the high localization of the thermal energy during the interface interaction; in this case, the rapid evaporation of the film occurs, the evaporated particles ionizing due to the interaction with the laser beam, leading to the formation of a plasma torch. The induced plasma significantly increases the absorption of the laser radiation, resulting in the efficient heating of the glass material. Another possible mechanism of bubble formation could be a phase explosion, that is characteristic for ablation processes.

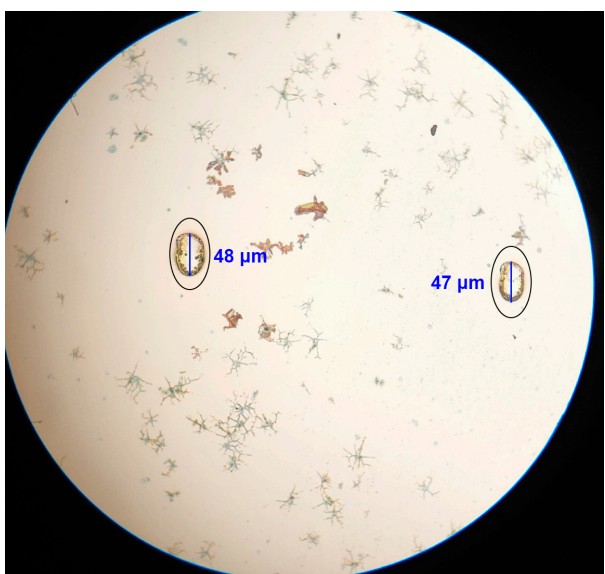

**Figure 6.** Evaporation of PVD glass coating during front plate irradiation.

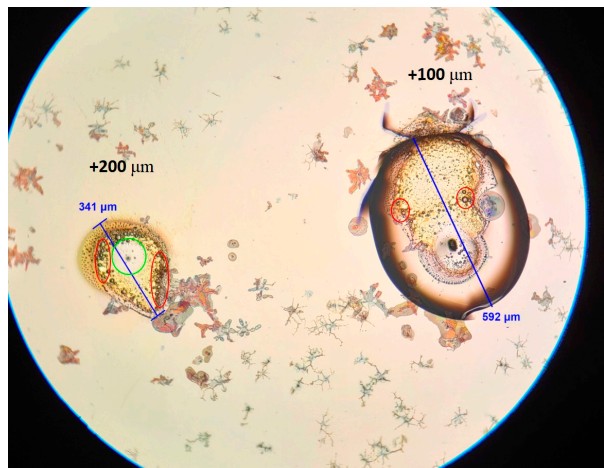

**Figure 7.** Structures induced on the glass surface df = +200 and +100 μm.

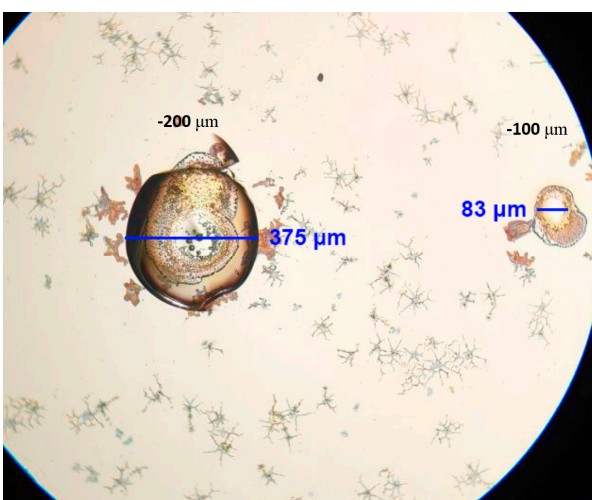

**Figure 8.** Structures induced on the glass surface df = −100 and −200 μm.

The most efficient bump formation was obtained for the following set of process parameters: P = 38 W, τ = 25 ms, f = 10 MHz, $E_{pulse}$ = 3.8 μJ, F = 0.39 J/cm². Figures 9–12 show a photograph of the formed bumps depending on the focus position.

It can be seen from the figures that the focus shift relative to the coating surface leads to an increase in the altered zone, but at the same time, the points are blurred; the points on Figure 11 have a lower height (~25 μm vs. ~50 μm), and in the case shown in Figure 12, bubble formation is not observed. The modified zone surface consists of incompletely evaporated coating. It is worth noting that microbubbles can be observed in the volume of structures formed on the surface; most likely, this is due to the intense explosive evaporation of the material and dissolution of the atmospheric gases in the melted material, which was rapidly quenched, so the gas bubbles were unable to escape the bump volume.

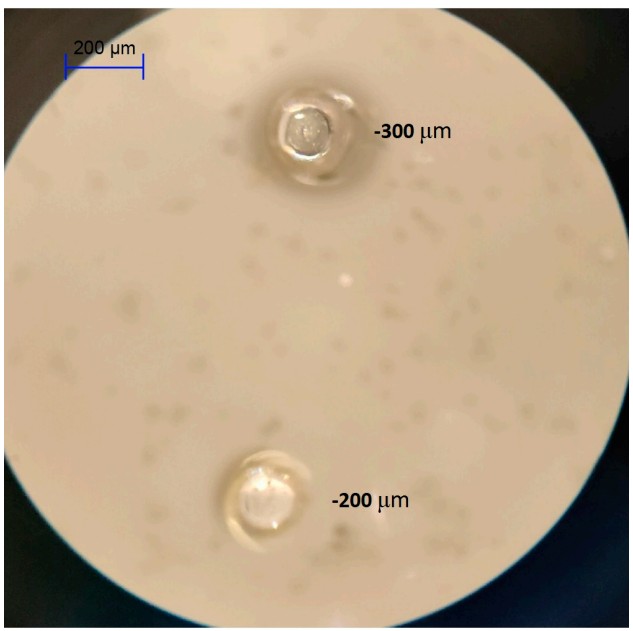

**Figure 9.** Bumps formed on the glass surface df = −300 and −200 μm.

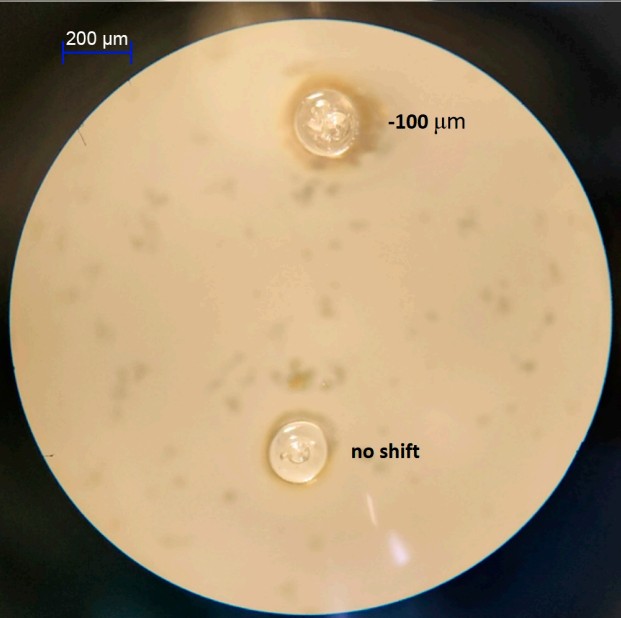

**Figure 10.** Bumps formed on the glass surface df = −100 and +0 μm.

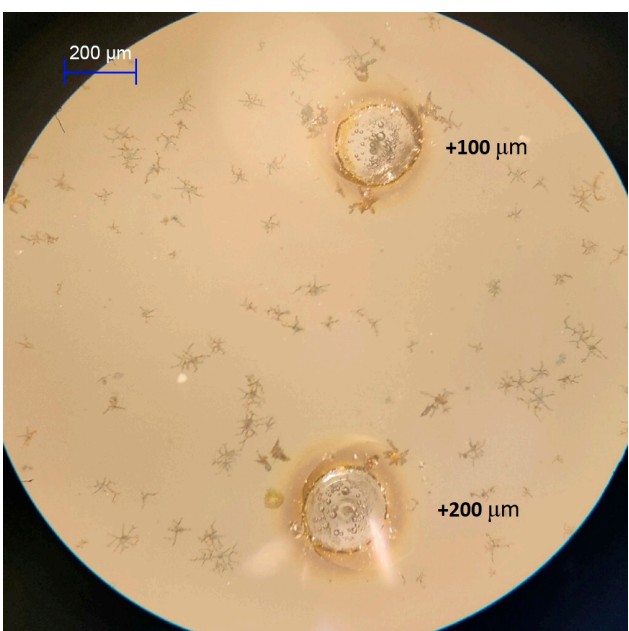

**Figure 11.** Bumps formed on the glass surface df = +100 and +200 µm.

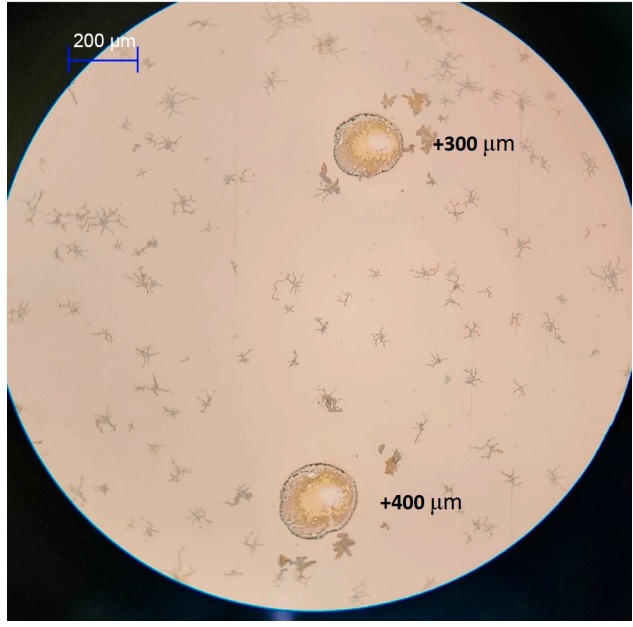

**Figure 12.** Bumps formed on the glass surface df = +300 and +400 µm.

A similar method was used for the FTO glasses; effective bump formation was achieved at $E_{pulse}$ = 3.2–3.8 µJ, F = 0.32–0.39 J/cm² (Figures 13 and 14), wherein, at pulse energy $E_{pulse}$ = 4 µJ (F = 0.41 J/cm²), intense crack formation was observed (Figure 15). This may be due to the higher absorption of FTO. One can also observe the splashing of the material near the impact area (blue circles), which indicates the rapid explosion-like evaporation of the coating and glass material. Thus, there is an upper limit of the pulse energy at which the effective formation of the bump can be achieved. This value also depends on the coating material, exposition time, and focusing parameters (laser fluence). In this case, the process window for FTO was established at pulse energy values $E_{pulse}$ = 2.7–3.6 µJ (F = 0.26–0.36 J/cm²), pulse repetition rate f = 10 MHz, and exposition time τ = 12.5–50 ms. Figures 14 and 15 show photographs of the bumps induced on the FTO glasses. Similar

to the PVD glass, the defocusing of the laser beam leads to coating evaporation without bubble formation, as shown on Figure 16.

The resulting height of the induced structures of 45–55 μm was measured with a mechanical micrometer. The height was also measured using contact profilometry; the height of the structures in this case was in good agreement with the data obtained when measured with a micrometer. Figure 17 shows a graph of the height of the structures obtained for the $E_{pulse}$ = 3.8 μJ, F = 0.39 J/cm$^2$, τ = 25 ms mode, depending on a small focus shift relative to the interface. Within this mode, the stable formation of structures that are uniform in height is possible, even when the focus is disrupted, which makes the process resistant to disturbances, such as vibration or thickness nonuniformities, which is essential for process scaling.

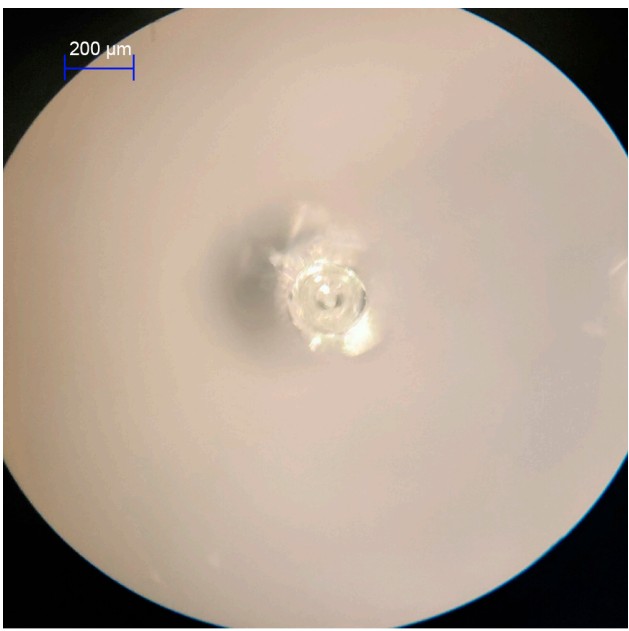

**Figure 13.** Structures induced on the FTO glass, focused directly on the interface.

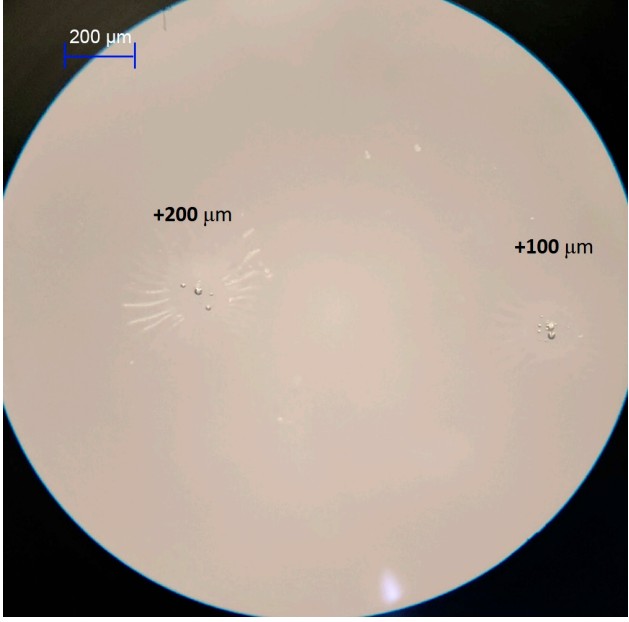

**Figure 14.** Structures induced on the FTO glass at focus position df = +200 and +100 μm.

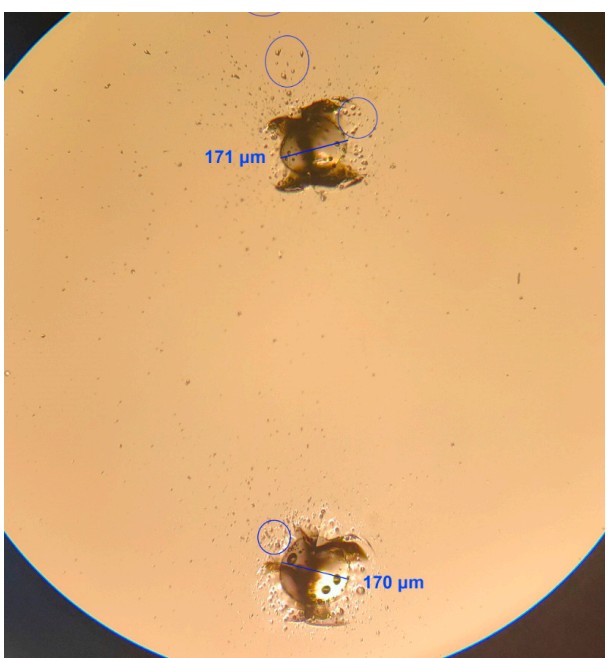

**Figure 15.** Cracking of FTO glass with parameters of $E_{pulse}$ = 4 μJ, F = 0.41 J/cm$^2$τ = 25 ms, f = 10 MHz.

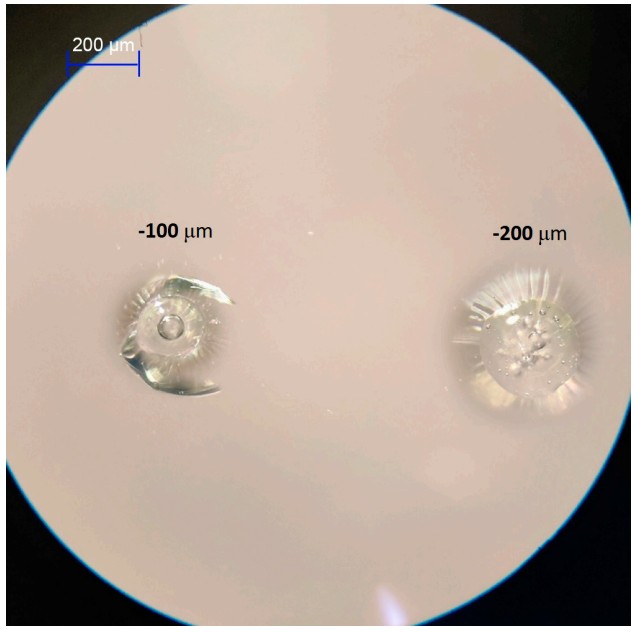

**Figure 16.** Structures induced on the glass surface df = −100 and −200 μm.

It can be seen from the profilogram that the bumps have a well-defined semi-spherical shape, which is confirmed by the photographs obtained with the microscope. One can observe a groove that surrounds the molded bumps. This may be due to the contraction of the melt during the formation of the spherical surface. It is also worth considering that the resolidification of the glass may be accompanied by a glass density change. It was found that within the framework of the effective bump formation method, the pulse energy has a fairly small effect on the height of the structures (within the bump formation method). It seems that the geometry of the bumps is mostly determined by the shape and size of the focal shape. At the same time, a shift in the focus position (>100 μm) leads to a drastic decrease in the height of the formed structure, caused by a decrease in the energy density and degradation of the energy profile. This significantly limits the ability to tune

the formation parameters in order to define the structure height. It was found that the most effective considered tuning parameter is the laser exposure time (number of pulses). Varying the exposure time, it became possible to obtain structures with heights of 8–12 and 20–25 μm. Figures 18 and 19 show photographs of the obtained structures alongside the corresponding profilograms.

One should note that the formation of cracks can be observed on the surface of structures 1 and 4, the presence of which can also be seen on the profilogram; however, this phenomenon cannot be associated with an increase in the pulse energy, since structure 2 does not have cracks on its surface, and in the case of structure 3, no defects are present. This phenomenon most likely has a fluctuational nature due to the nonequilibrium of the fast process of bump formation; the difference may also be due to the local features of the sample or coating thickness.

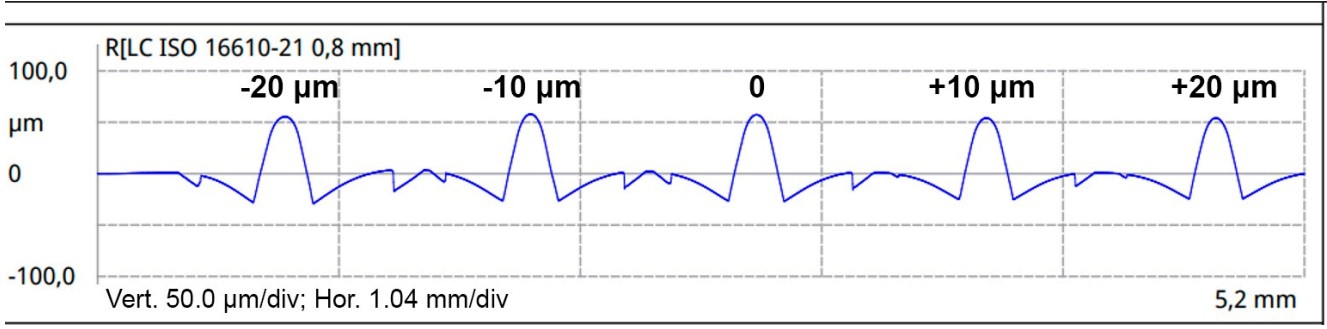

**Figure 17.** Profile of bump heights depending on focusing disturbance (small focal shift).

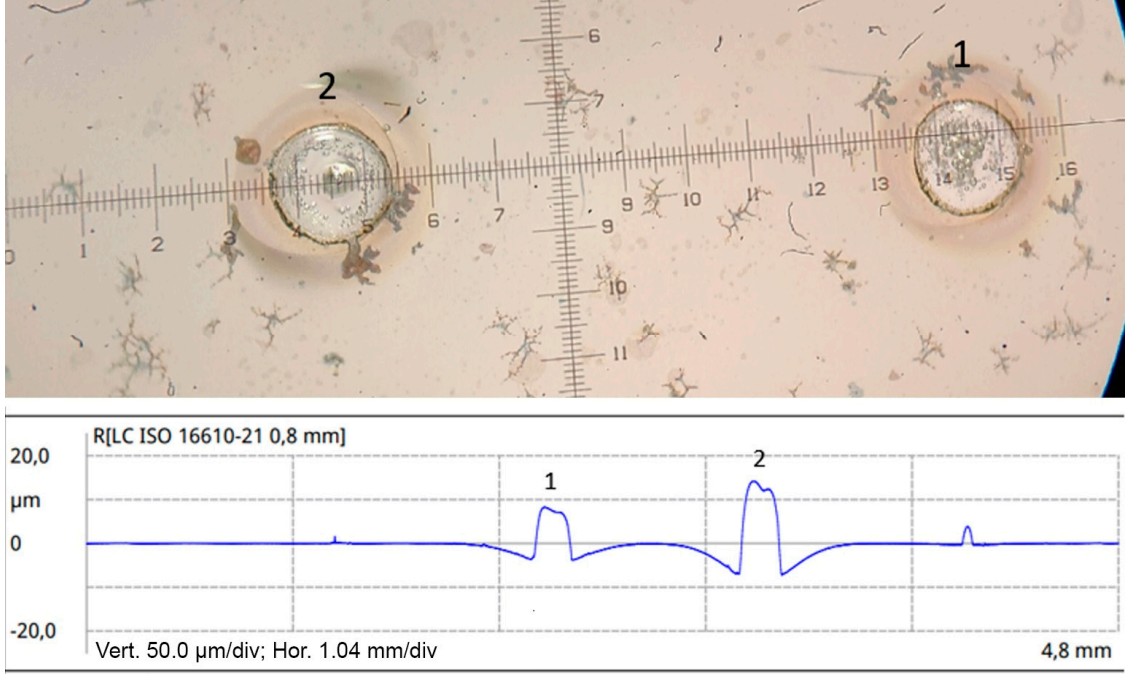

**Figure 18.** Bumps formed on PVD glass: 1—$E_{pulse}$ = 3.2 μJ (F = 0.32 J/cm$^2$), $\tau$ = 10 ms: 2—$E_{pulse}$ = 3.3 μJ (F = 0.32 J/cm$^2$), $\tau$ = 12.5 ms.

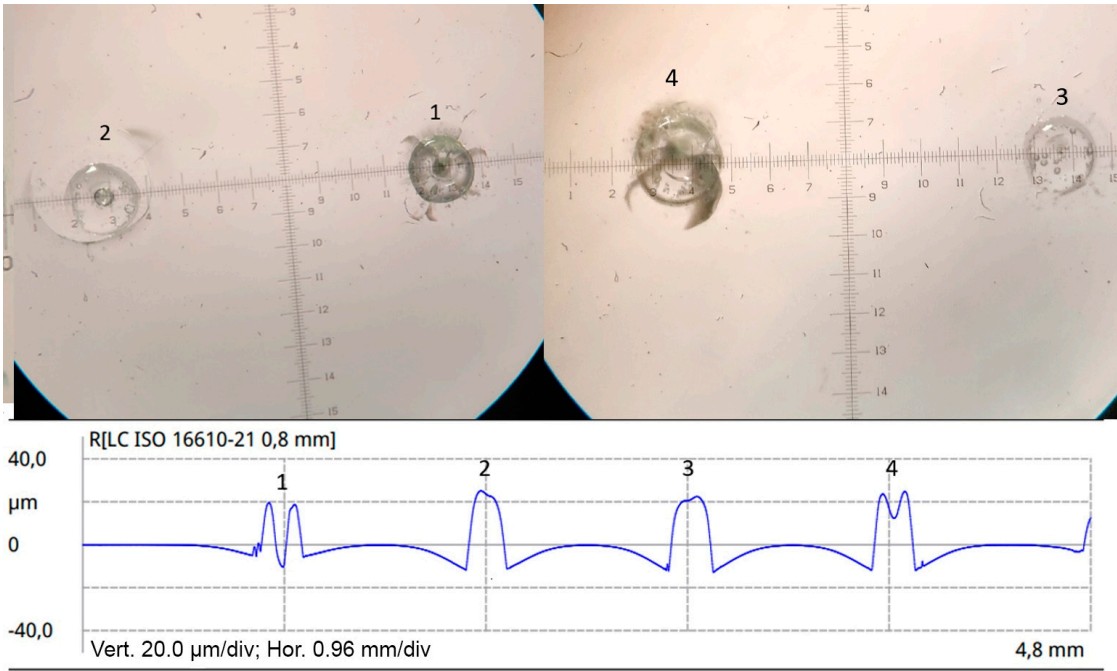

**Figure 19.** Bumps formed on FTO glass at $\tau$ = 10 ms: 1—$E_{pulse}$ = 3.0 $\mu$J (F = 0.30 J/cm$^2$); 2—$E_{pulse}$ = 3.2 $\mu$J (F = 0.32 J/cm$^2$); 3—$E_{pulse}$ = 3.3 $\mu$J (F = 0.33 J/cm$^2$), 4—$E_{pulse}$ = 3.4 $\mu$J (F = 0.34 J/cm$^2$).

To summarize, during the experiments, we have identified the following characteristic features and modes of interaction of the picosecond pulses with the glass–conductive coating interface depending on the process parameters and laser exposure geometry:

Volume modification: when the laser pulses were applied to the glass without a coating, it was found that the energy density was not enough to activate the processes of melting the glass material near the surface; however, it was found that at a high value of the pulse energy, it is possible to observe the formation of altered areas in the volume, characteristic for the volumetric modification of the glass with ultrafast laser pulses.

Laser heating: the significant absorption of laser radiation by the coating (~80% FTO and ~15% PVD i/iM-glass) provides an effective channel for the heating and melting of the film material, wherein the glass surface is not affected due to the higher melting point. This effect is observed at a low pulse energy.

Film evaporation: When the ablation threshold value is reached, the process of the evaporation of the coating from the glass surface begins, while the glass surface remains practically unaffected. The energy fluence required for evaporation lies in the range of 0.10–0.20 J/cm$^2$.

Surface melting: When the density of the surface energy reaches F = 0.20–0.24 J/cm$^2$, the process of glass melting is initiated. This is due to the intensification of the thermal effects at the interface caused by the absorption of radiation by plasma induced on the surface, formed due to the field ionization of the coating vapor. In this case, one can observe limited surface mixing due to the displacement of the melt by recoil vapor.

Bump formation: Reaching the threshold energy value activates the process of the rapid formation of a melt droplet on the surface. When the laser exposure is removed, the melted materials solidify, forming a spherical structure on the surface. In this case, two mechanisms may be involved in the process of bump formation:

a.   The activation process of the viscous flow of the glass melt upon reaching the glass transition temperature. A similar process is observed when structures are formed on the surface of the glass using FIR (10.6 $\mu$m) radiation [35].

b. Phase explosion caused by the rapid heating and evaporation of the glass in the focal volume, followed by the fast solidification of the melt with different (high or low temperature) glass modifications. Such a process is typical for ultrafast laser pulses' interaction with opaque materials [36].

It seems that both of these processes are involved in bump formation; however, taking into account the rapid nature of the interaction, the processes of the directed flow of molten material cannot be sufficiently effective. At the same time, the high peak power of picosecond laser radiation ensures the rapid boiling and evaporation of the material, necessary for a phase explosion near the surface. Thus, we assume that the key mechanism for the formation of bumps is a phase explosion in the surface region of the glass.

Bump elimination: At a high energy density, the destruction of formed structures is observed and manifests itself as the "bursting" of bubbles formed on the surface. In this case, phase explosion is powerful enough to break the bubble structure, resulting in the formation of a round crater on glass surface.

Table 1 shows list of all the featured processes alongside the threshold energy density values and photographs of the glass surface. All data were obtained at the following process parameters: f = 10 MHz, number of pulses n $\approx 10^5$, focused beam diameter $d_f$ = 50 µm. It is worth noting that the heating and evaporation thresholds differ significantly for the FTO and i/iM-glass, wherein for glass melting, the bump formation and elimination values of the threshold are almost the same. This can be attributed to the fact that the processes of the melting and evaporation of the coating is dependent on the physical and thermal properties of the coating material, while glass melting and bump formation are determined by the glass's properties. In that case, an ionized coating material is highly absorbing for both coatings; therefore, the influence of the film composition is neglectable.

**Table 1.** Characteristic processes.

| Process | Surface Energy Density (F), mJ/cm$^2$ | Irradited Area |
|---|---|---|
| Laser heating and melting of coating | 90 ± 50 (FTO)<br>190 ± 50 (i/iM-glass) |  |
| Coating evaporation | 160 ± 20 (FTO)<br>240 ± 20 (i/iM-glass) |  |

**Table 1.** *Cont.*

| Process | Surface Energy Density (F), mJ/cm$^2$ | Irradited Area |
|---|---|---|
| Glass surface melting | $(200 \div 220) \pm 20$ (FTO)<br>230–250 $\pm$ 20 (i/iM-glass) |  |
| Bump formation | $250 \pm 20$ (FTO)<br>$270 \pm 20$ (i/iM-glass) |  |
| Bump elimination | $400 \pm 20$ |  |
| Volume Modification | >300 |  |

## 4. Conclusions

Smart windows are an essential technology for energy-saving applications, which is nowadays vital due to global warming, the depletion of energy resources, and the need for improving energy efficiency. In particular, the problem of heating residential or office premises has recently become acute, which can be partly solved by the application of smart windows. However, to achieve a noticeable impact, it is necessary to push for the widespread introduction of this technology, which requires the development of cost-effective and high throughput manufacturing process. To achieve this, the process of laser bump formation using 50 ps laser radiation was studied. As a result, the process of the

formation of spherical structures on the surface of glasses with two types of coatings (FTO and i/iM-glass) was implemented. The parameter window for effective bump formation was determined, characterized by the following parameters: pulse repetition frequency f = 10 MHz, exposition time $\tau$ = 10–50 ms, pulse energy F = 0.31–0.40 J/cm$^2$ for i/iM-glass coating and F = 0.27–0.36 J/cm$^2$ for FTO. Those process parameters have allowed for the obtainment of bump formation with heights up to 50–55 μm, while the size of the structures can be altered by tuning the formation parameters down to 10–20 μm. The diameters of the structure was within the range of 150–250 μm and was determined mainly by the shape of the focused laser beam. However, the problems of stress formation and the cracking of bumps under loading still to be solved.

**Author Contributions:** Conceptualization, S.I. and G.M.; Methodology, S.I. and G.M.; Formal analysis, S.I.; Investigation, S.I.; Writing—original draft, S.I.; Writing—review & editing, S.I., A.P. and G.M.; Supervision, A.P.; Project administration, A.P. All authors have read and agreed to the published version of the manuscript.

**Funding:** This research was funded by the Ministry of Science and Higher Education of the Russian Federation as a part of the World-class Research Center program: Advanced Digital Technologies (contract No. 075-15-2022-311 dated 20 April 2022).

**Data Availability Statement:** Data are contained within the article.

**Conflicts of Interest:** The authors declare no conflict of interest.

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
