# Peer review of "Picosecond Laser-Induced Bump Formation on Coated Glass for Smart Window Manufacturing"

_jmmp, doi:10.3390/jmmp8010001_

Round 1
Reviewer 1 Report
Comments and Suggestions for Authors
The paper in general is well written, conveyed the objectives clearly. There are a few areas that this reviewer would like to request from the authors. The comments are in the attached file. Thank you

Author Response
Points 1-6 are now covered in more detail - lines 94-132 (highlighted).
7. System jitter is described - lines 152-153
8. Corrected
9. The beam profile, beam quality is now mentioned (lines 137-139), the focusing procedure is now described in detail (lines 159-166)
10. Elliptical shape is explained (lines 206-210)
11. Points are now labeled corresponding to the focal shift
12. The energy fluence are now presented alongside with pulse energies
13. Ra, Rz, Rmax - roughness parameters, which has no practical value to the subject of paper, so it was removed
14. The laser used in the experiments has a fixed pulse duration that can't be tuned, the pulse duration value is given in the line 135
15. We are working on the industrial deployable system at the moment and unfortunately we are unable to show it yet. We hope that we would present it in upcoming works.
Thank you very much!
Reviewer 2 Report
Comments and Suggestions for Authors
The aim of this work was to study interaction of picosecond pulsed laser with coated glass surface. The authors investigation shown that glass bump formation is influenced by different laser parameters, like pulse repetition frequency, exposition time, pulse energy etc.
My general conclusion is that manuscript should be accepted for publication in Journal of Manufacturing and Materials Processing after minor revision.
Comments and suggestions:
1. Page 2, lines 69 and 78: There is no need to refer to Fig 1. two times.
2. Page 3-4, Lines 117-120: If your laser system is further described in some of your previous papers, that paper should be referenced.
3. Page 5, Figure 3, and Page 7, line 203: It would be beneficial to also give corresponding fluence, together with other already stated parameters.
4. Page 8, Figure 11: typing, correct forxmed into formed.
5. Page 10, Figure 17: The raw with focus shift should be labeled (i guess marks -20µm, -10 µm, 0, +10 µm, +20µm represent focus shift, but it must be clear). Also, It is hard to notice any height difference between profiles and in manuscript it is explained that “ stable formation of structures that are uniform in height is possible, even when the focus is disrupted, which makes the process resistant to disturbances“. This means that bump height is almost independent from small changes of focal shift, making text under Figure 17 slightly misleading. Figure 17 text should be changed to better describe the process.
6. Page 12, line 281: Energy value [µJ] for initialization of glass melting process is given, not energy density. Please give laser fluence value.
Author Response
- Corrected
- We haven't described our system yet
- The energy fluence is now provided alongside with pulse energy
- Corrected
- The wording in figure description has been changed to avoid possible confusion
- Corrected
Thank you!
Reviewer 3 Report
Comments and Suggestions for Authors
Smart-window glass is an interesting topic in recent years and the fabrication and micro machining may attract attentions from industry.
The methodology by ultrafast IR laser presented is widely used and sound. However, the merit of the research should be elucidated since pico-second IR laser may not the only solution for micro machining of glass. Also, the author should explain the material properties of smart-window glass will affect the results from laser irradiation. One of the issues cited is backside ablation is quite normal in laser machining process in transparent materials which should be taken care of. The significance of the research should be clear to the readers.
Author Response
We tried our best to to expand and deepen the rationale in the first part of the work. (Lines 97-142, highlighted).
Thank you for your help!
Round 2
Reviewer 3 Report
Comments and Suggestions for Authors
The amended supplement helps a lot.